# Prognostic significance of SHP2 (PTPN11) expression in solid tumors: A meta-analysis

**Jiupeng Zhou**[1]*, **Hui Guo**[1,2], **Yongfeng Zhang**[1], **Heng Liu**[1], **Quanli Dou**[1]

**1** Xi'an Chest Hospital, Xi'an, Shaanxi Province, China, **2** The First Affiliated Hospital of Xi'an Jiaotong University, Shaanxi Province, China

* 44996323@qq.com

## Abstract

### Background

SHP2 is a latent biomarker for predicting the survivals of solid tumors. However, the current researches were controversial. Therefore, a meta-analysis is necessary to assess the prognosis of SHP2 on tumor patients.

### Materials and methods

Searched in PubMed, EMBASE and web of science databases for published studies until Jun 20, 2021. A meta-analysis was performed to evaluate the affect of SHP2 in clinical stages, disease-free survival (DFS) and overall survival (OS) in tumor patients.

### Results

This study showed that the expression of SHP2 had no significant correlation with clinical stages (OR: 0.91; 95% CI, 0.60–1.38; P = 0.65), DFS (HR = 0.88; 95%CI: 0.58–1.34; P = 0.56) and OS (HR = 1.07, 95%CI: 0.79–1.45, P = 0.67), but the prognostic effect varied greatly with tumor sites. High SHP2 expression was positively related to early clinical stage in hepatocellular carcinoma, not associated with clinical stage in the most of solid tumors, containing laryngeal carcinoma, pancreatic carcinoma and gastric carcinoma, etc. Higher expression of SHP2 could predict longer DFS in colorectal carcinoma, while predict shorter DFS in hepatocellular carcinoma. No significant difference was observed in DFS for non-small cell lung carcinoma and thyroid carcinoma. Higher SHP2 expression was distinctly related to shorter OS in pancreatic carcinoma and laryngeal carcinoma. The OS of the other solid tumors was not significantly different.

### Conclusions

The prognostic value of SHP2 might not equivalent in different tumors. The prognostic effect of SHP2 is highly influenced by tumor sites.

**Data Availability Statement:** All relevant data are within the paper and its Supporting Information files.

**Funding:** The funders had no role in study design, data collection and analysis, decision to publish, or preparation of the manuscript.

**Competing interests:** The authors have declared that no competing interests exist.

## Introduction

According to the 2018 National Cancer Report, there would be 1 735 350 new tumor cases and 609 640 tumor related deaths [1]. Although great progress has been made in diagnosis and treatment, the therapeutic effects of most tumors are still disappointing [2]. Considering this situation, more and more researchers begin to look for ideal indicators that can predict the prognosis of tumor.

Src homologous phosphotyrosine phosphatase 2 (SHP2), encoded by PTPN11, is widely expressed in cells, which promotes cell proliferation and movement [3,4]. SHP2 is an intracellular tyrosine phosphatase with two tandem repeat SRC homologous 2 domains. It is the main regulator of tyrosine kinase receptor, cytokine receptor and hormone signal transduction [5,6]. It is initially found that PTPN11 mutation induces SHP2 activation, leading to Noonan syndrome and juvenile leukemia [7,8].

The recent studies discovered that SHP2 was a potential biomarker for the prognosis of solid tumors [9–19]. However, certain studies were controversial. Some studies showed that high SHP2 expression might have a bad effect on the prognosis of tumor patients [9–11]. Others found that high SHP2 expression had nothing to do with the poor prognosis of tumor patients [12–15]. So far as to certain studies considered that high SHP2 expression might be related to the good prognosis of tumor patients [16–19]. Hence, the objective of this meta-analysis is to assess the prognostic significance of SHP2 expression in tumor patients from the possible existence of deviations.

## Materials and methods

### Literature search strategy

Embase, PubMed, web of science and other website databases were comprehensively searched up to Jun 20, 2021. The search keywords and search strategy were as below: "PTPN11", "SHP2", "SH-PTP2", "PTP2C", "BPTP3", "cancer", "tumor", "clinicopathology", "prognosis", and "survival". We checked up the reference documents of the retrieved literature to refrain from leaving out relevant studies, too. Besides, references listed at the end of the relevant reviews were all conducted manually to identify potential usable studies.

### Study selection

The included criteria were as below: 1) SHP2 expression detected in primary cancer tissues; 2) patients split into two groups according to the expression of SHP2; 3) clinicopathological parameters, disease-free survival (DFS)/progression-free survival (PFS) or overall survival (OS) were provided; 4) enough data to collect. The articles, letters, or experiments on animal models and iterated research publications were removed.

### Date extraction and quality evaluation

Data were extraced by two investigators on their own, as was the quality assessment. If there were differences, they would be solved through panel discussion. The information and data were collected from every study in the form of a specific design: the author, year of publication, the country, type of cancer, patient number, patients in high SHP2 expression group and low SHP2 expression, patients with TNM (I-II/III-IV), follow-up data, and cutoff value of SHP2. For OS, PFS/DFS, the risk ratio (HR) and relevant 95% confidence interval (CI) were immediatly collected from the primary studies. If HR and 95% CI were not specified in the studies, they were evaluated with the means described by Tierney et al [20] and Parmar et al [21]. If both multivariate analysis and univariate analysis were used to evaluate OS, the HR and

relevant 95% CI derived from the multivariate analysis were used. The quality of each study was appraised with Newcastle-Ottawa scale (NOS). The NOS score varied from 0 to 9. If NOS score was 6 or more, the research was supposed to be of high quality.

## Statistical analysis

Stata SE13.0 software and Revman5.3 software were used for meta-analysis. The prognosis (such as PFS/DFS, OS) was assessed by HR with corresponding 95%CI. For binary variables, odds ratio (OR) and corresponding 95%CI were used. The heterogeneity of the included studies was measured by $I^2$ and Q statistics. The P value <0.05 and $I^2$ >50% were considered severe heterogeneity. The fixed-effects model was selected if there was no obvious heterogeneity among included studies (P>0.05, $I^2$<50%). If not, the random-effects model was used (P≤0.05, $I^2$≥50%). Concurrently, subgroup analysis was performed to further explore the effect of SHP2 expression on prognosis. Publication bias was assessed by Egger's and Begg's funnel plot test. P<0.05 was statistically significant.

## Results

### The literature search and selection

After the preliminary search algorithm, 1964 articles were retrieved. Through the title and abstract, the irrelevant articles were excluded. That 168 articles were evaluated. Literature of review article, case report, or without survival datas, binary variables and valuable datas were excluded. Ultimatly, 15 articles were identified to further evaluate in this meta-analysis, involving 2897 patients (Fig 1).

### Characteristics of included studies

In 15 studies, the average sample size for every study was 206.9 cases (range: 17–347). Eleven of them were conducted in China, two in Korea, and two in Spain. Eight different cancer types were included in this meta-analysis, including three of gastric carcinoma, three of hepatocellular carcinoma, two of laryngeal carcinoma, two of thyroid carcinoma, one of pancreatic cancer, two of esophagus cancer, one of colorectal cancer, two of non-small cell lung cancer, and colorectal cancer. SHP2 expression was detected by immunohistochemistry in twelve and by real-time PCR in three. There were 1339 patients in high SHP2 expression group and 1558 patients in low SHP2 expression one. Six studies reported on the association of SHP2 expression and clinical stage. Eleven articles covered the relationship between SHP2 expression and OS. Five studies evaluated the association of SHP2 expression with DFS/PFS (Tables 1 and 2).

### Main analysis

On meta-analysis of eleven studies assessing the TNM stage, SHP2 expression was not associated with clinical stages (OR: 0.91; 95% CI, 0.60–1.38; P = 0.65). However, considerable heterogeneity was observed in different studies ($I^2$ = 64%; P = 0.002) (Fig 2). Five studies discussed the relationship between SHP2 and DFS. The results displayed that SHP2 expression was not distinctly related to DFS (HR = 0.88; 95%CI: 0.58–1.34; P = 0.56), with a great heterogeneity ($I^2$ = 63%; P = 0.03) (Fig 3). Thirteen articles covered 333 patients with OS in the light of SHP2 expression. The analysis showed a pooled HR value (HR = 1.07, 95%CI: 0.79–1.45, P = 0.67), with a great heterogeneity ($I^2$ = 72%; P<0.001) (Fig 4).

 **PRISMA 2009 Flow Diagram**

**Identification**

Records identified through
database searching
(n=1964)

Additional records identified
through other sources
(n=33)

Records after duplicates removed
(n=997)

**Screening**

Records screened
(n=168)

Records excluded
(n=829)

**Eligibility**

Full-text articles assessed
for eligibility
(n=23)

Full-text articles excluded,
with reasons
(n=145)

**Included**

Studies included in
quantitative synthesis
(meta-analysis)
(n=15)

*From:* Moher D, Liberati A, Tetzlaff J, Altman DG, The PRISMA Group (2009). *P*referred *R*eporting *I*tems for *S*ystematic Reviews and *M*eta-*A*nalyses: The PRISMA Statement. PLoS Med 6(7): e1000097. doi:10.1371/journal.pmed1000097

**For more information, visit www.prisma-statement.org.**

**Fig 1. A flowchart describing the procedures of document retrieval and selection.**

## Subgroup meta-analyses

Subgroup analysis was performed in view of the great heterogeneity. When these studies were grouped according to tumor sites, the heterogeneity decreased significantly. High SHP2 expression was positively correlated with early clinical stage in hepatocellular carcinoma (OR: 0.4; 95% CI, 0.24–0.66; P<0.001). SHP2 expression was not associated with clinical stage in laryngeal carcinoma (OR: 2.75; 95% CI, 0.14–55.17; P = 0.52), pancreatic carcinoma (OR: 1.45; 95% CI, 0.58–3.62; P = 0.43), gastric carcinoma (OR: 1.23; 95% CI, 0.85–1.78; P = 0.27), esophagus carcinoma (OR: 0.98; 95% CI, 0.4–2.39; P-0.97), colorectal carcinoma (OR: 0.84; 95% CI, 0.48–1.45; P = 0.53) and thyroid carcinoma (OR: 0.69; 95% CI, 0.38–1.26; P = 0.23) (Fig 5). When subgroup analysis was carried on the relationship of SHP2 and DFS, the results indicated that higher expression of SHP2 could predict longer DFS in colorectal carcinoma (HR: 0.45; 95% CI, 0.23–0.88; P = 0.019), while predict shorter DFS in hepatocellular carcinoma (HR: 1.38; 95% CI, 1.01–1.88; P<0.05). No significant difference was observed in DFS for non-small cell lung carcinoma (HR: 0.92; 95% CI, 0.58–1.46; P = 0.72) and thyroid carcinoma (HR: 0.75; 95% CI, 0.42–1.36; P = 0.075) (Fig 6). Subgroup analysis displayed that higher expression of SHP2 was significantly associated with shorter OS in pancreatic carcinoma (HR: 1.98; 95% CI, 1.17–3.37; P<0.001) and laryngeal carcinoma (HR: 2.84; 95% CI, 1.20–6.74; P = 0.02). There was no obvious difference in OS for colorectal carcinoma (HR: 1.62; 95% CI, 0.89–2.95; P = 0.12), gastric carcinoma (HR: 1.23; 95% CI, 0.93–1.62; P = 0.14), non-small cell lung cancer (HR: 1.15; 95% CI, 0.67–1.98; P = 0.62), thyroid carcinoma (HR: 1.11; 95% CI, 0.28–4.35; P = 0.88), hepatocellular carcinoma (HR: 0.88; 95% CI, 0.69–1.12; P = 0.29) and esophagus carcinoma (HR: 0.66; 95% CI, 0.38–1.14; P = 0.14) (Fig 7).

**Table 1. The basic information and data of all included studies in the meta-analysis.**

| Author(year) | Country | Cancer type | Total number | PTPN11expression | | Detection method | Criterion of high expression | Quality stars (NOS) |
|---|---|---|---|---|---|---|---|---|
| | | | | High | Low | | | |
| Jin Soo Kim 2009 | Korea | GC | 92.000 | 78 | 14 | IHC | The cells stained≥30% | 7 |
| Chengying Jiang2012 | China | HCC | 333.000 | 62 | 271 | IHC | H-score≥80 | 8 |
| L.B. Dong2013 | China | LC | 17.000 | 15 | 2 | IHC | IRS≥2 | 7 |
| Jing Jiang2013 | China | GC | 305.000 | 235 | 70 | IHC | H-score≥100 | 9 |
| JIA GU2014 | China | LC | 112 | 56 | 56 | IHC | | 7 |
| Tao Han2015 | China | HCC | 301.000 | 150 | 151 | IHC | ≥the median score | 8 |
| ZHONG-QIANHU2015 | China | TC | 65 | 41 | 14 | IHC | H-score≥200 | 7 |
| Jiawei Zheng2016 | China | PC | 79.000 | 44 | 35 | IHC | IRS≥4 | 9 |
| Chen Qi2017 | China | EC | 76 | 33 | 43 | IHC | ≥the median score | 7 |
| Yan Huang2017 | China | CC | 270.000 | 126 | 144 | IHC | IRS≥9 | 9 |
| Jun Cao 2018 | China | TC | 313.000 | 180 | 113 | IHC | IRS≥5 | 8 |
| Min-Kyung Kim2018 | Korea | HCC | 50.000 | 29 | 21 | IHC | ≥10% | 7 |
| Niki Karachaliou2019 | Spain | NSCLC | 47.000 | 24 | 23 | Real-time PCR | | 7 |
| Ivan Macia2020 | Spain | NSCLC | 102.000 | 49 | 53 | Real-time PCR | ≥2-fold | 7 |
| Jing Chen2020 | China | GC | 347 | 86 | 261 | Real-time PCR | ≥the third quartile | 9 |
| Jing Chen2020 | China | EC | 115 | 27 | 88 | Real-time PCR | ≥the third quartile | 9 |
| Jing Chen2020 | China | CRC | 273 | 74 | 199 | Real-time PCR | ≥the third quartile | 9 |

GC, gastric carcinoma; HCC, hepatocellular carcinoma; LC:laryngeal carcinoma; TC,thyroid carcinoma; PC, pancreatic carcinoma; EC,esophagus carcinoma; CC, colorectal carcinoma; NSCLC, non-small cell lung carcinoma.

**Table 2. The research results of all included studies in the meta-analysis.**

| Author (year) | PTPN11 expression | TNM stage | | OS | | | | DFS | | | |
|---|---|---|---|---|---|---|---|---|---|---|---|
| | | I/II | III/IV | HR | 95%CI | In (HR) | Se (InHR) | HR | 95%CI | In (HR) | Se (InHR) |
| Jin Soo Kim 2009 | High | 43 | 35 | | | | | | | | |
| | Low | 11 | 3 | | | | | | | | |
| Chengying Jiang2012 | High | 38 | 24 | 0.460 | 0.300–0.710 | -0.770 | 0.220 | | | | |
| | Low | 105 | 166 | | | | | | | | |
| L.B. Dong2013 | High | 4 | 11 | | | | | | | | |
| | Low | 1 | 1 | | | | | | | | |
| Jing Jiang2013 | High | 60 | 175 | 1.060 | 0.700–1.610 | 0.060 | 0.210 | | | | |
| | Low | 15 | 55 | | | | | | | | |
| Jia Gu2014 | High | | | 2.837 | 1.196–6.728 | 1.043 | 0.441 | | | | |
| | Low | | | | | | | | | | |
| Tao Han2015 | High | | | 1.393 | 1.021–1.899 | 0.331 | 0.158 | 1.370 | 1.010–1.870 | 0.320 | 1.160 |
| | Low | | | | | | | | | | |
| Zhong Qianhu2015 | High | 27 | 24 | | | | | | | | |
| | Low | 12 | 2 | | | | | | | | |
| Jiawei Zheng2016 | High | 15 | 29 | 2.045 | 1.168–3.367 | 0.685 | 0.270 | | | | |
| | Low | 15 | 20 | | | | | | | | |
| Chen Qi2017 | High | | | 0.730 | 0.340–1.580 | -0.310 | 0.390 | | | | |
| | Low | | | | | | | | | | |
| Yan Huang2017 | High | | | | | | | 0.447 | 0.227–0.877 | -0.807 | 0.345 |
| | Low | | | | | | | | | | |
| Jun Cao 2018 | High | | | 1.109 | 0.283–4.351 | 0.104 | 0.697 | 0.754 | 0.417–1.363 | -0.283 | 0.302 |

## Publication bias

Publication bias was evaluated by Egger's test. There was no publication bias for clinical stages (P = 0.477), DFS (P = 0.416), OS (P = 0.671) from the studies (Table 3).

## Discussion

SHP2 was initially believed as a proto-oncogene with acquired functional mutation in leukemia, which could activate hematopoietic stem cells (HSCs) and induce leukemia [22,23]. Then later on SHP2 was found to be overexpressed in some solid tumors including NSCLC [13] and

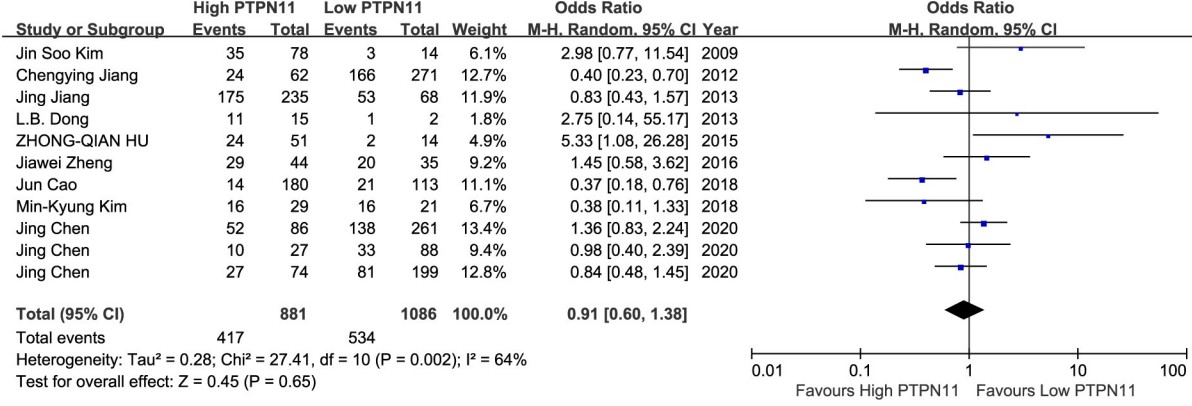

**Fig 2. A forest plot for the association between SHP2 expression levels with clinical stage.**

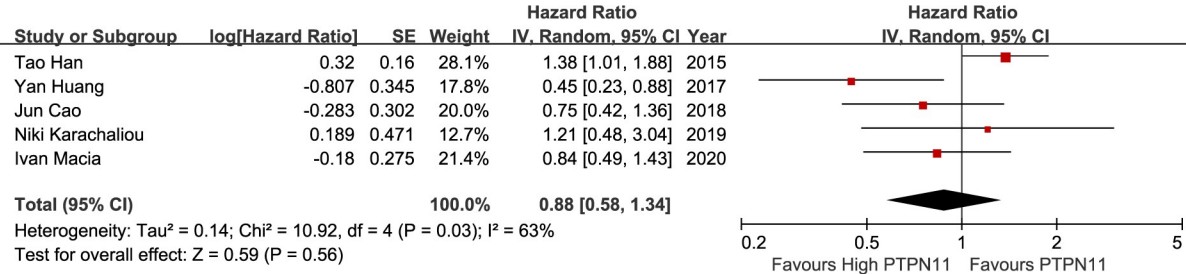

**Fig 3. A forest plot for the association between SHP2 expression levels with DFS.**

gastric cancer [12,24]. Meanwhile, SHP2 has been proved to be down regulated in hepatocellular carcinoma, inhibiting the development of hepatocellular carcinoma [18]. Because SHP-2 has important biological characteristics in tumor cells, studies on SHP-2 prognostic affect on several types of tumors have been carried out. However, the results were controversial. This study would define the significance of SHP-2 expression on the prognosis of patients with solid tumors.

This meta-analysis suggested that high SHP2 expression was positively correlated with early clinical stage in hepatocellular carcinoma, not associated with clinical stage in laryngeal carcinoma, pancreatic carcinoma, gastric carcinoma, esophagus carcinoma, colorectal carcinoma, and thyroid carcinoma. The higher expression of SHP2 predicted longer DFS in colorectal carcinoma, while predicted shorter DFS in hepatocellular carcinoma. No significant difference was observed in DFS for non-small cell lung carcinoma and thyroid carcinoma. Subgroup analysis displayed that higher SHP2 expression was distinctly related to shorter OS in pancreatic carcinoma and laryngeal carcinoma. There was no obvious difference in OS for colorectal carcinoma, gastric carcinoma, non-small cell lung cancer, thyroid carcinoma, hepatocellular carcinoma, and esophagus carcinoma. The results of this meta-analysis should be reliable given the high quality of the included studies whose NOS scores were ≥7 (S1 Table). These studies provided almost all the available evidences worldwide on the role of SHP2 in the prognosis of solid tumors. A advantage of bringing together worldwide evidences on the association between SHP2 and the prognosis of solid tumors was that there were large numbers of cases to assess reliably whether the association varies by tumour subtype. We found that high SHP2 expression was associated with shorter OS in two subtypes of solid tumors of pancreatic

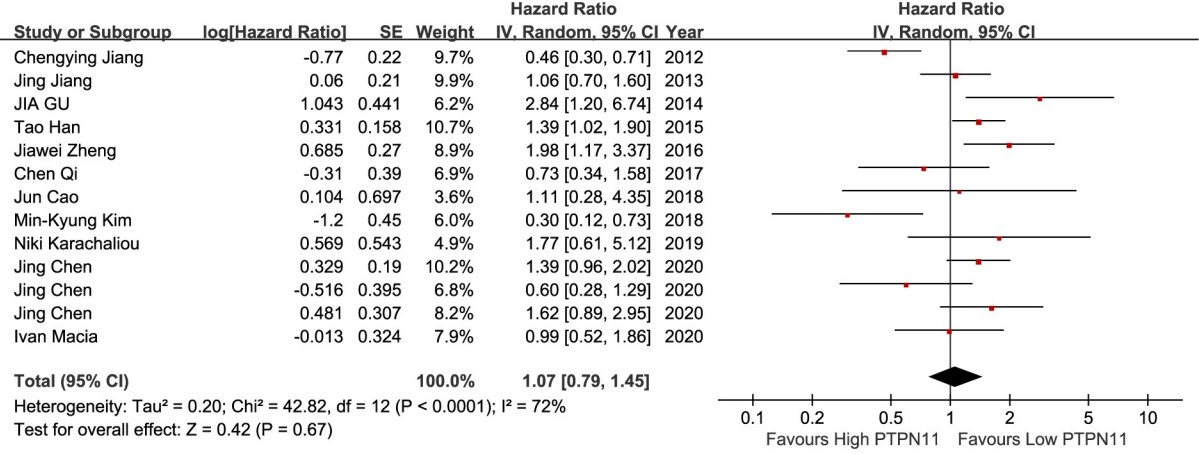

**Fig 4. A forest plot for the association between SHP2 expression levels with with OS.**

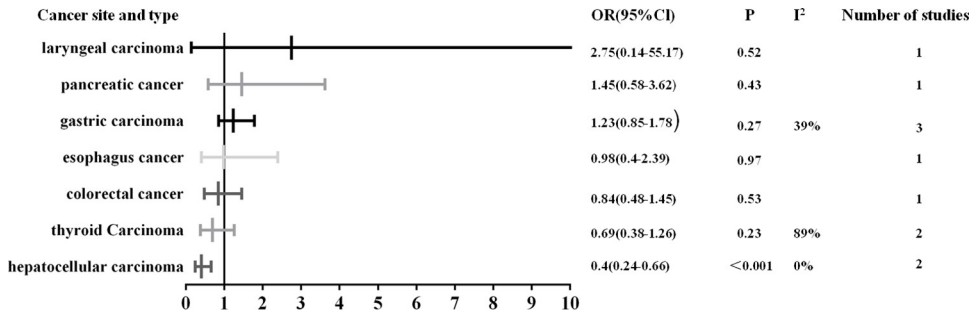

**Fig 5. Summary risk estimates of clinical stage by cancer sites.**

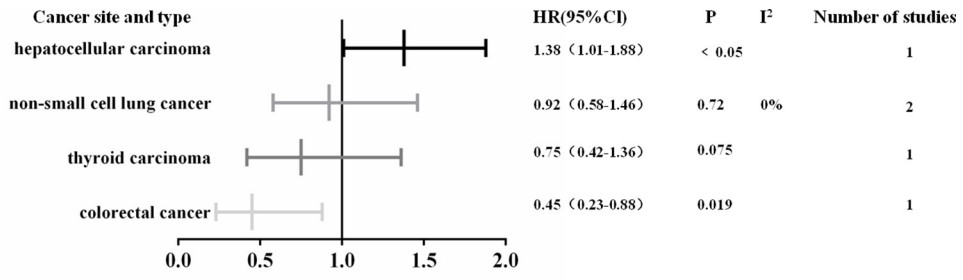

**Fig 6. Summary risk estimates of DFS by cancer sites.**

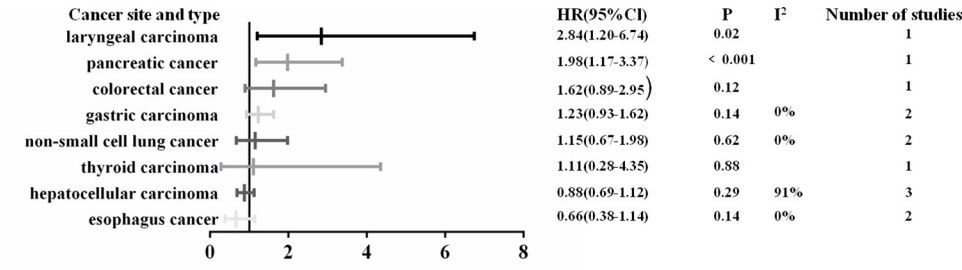

**Fig 7. Summary risk estimates of OS by cancer sites.**

**Table 3. The publication bias test including literatures.**

|       | Coef   | 95%CI         | t     | P     |
|-------|--------|---------------|-------|-------|
| TNM   | 0.607  | -1.247,2.463  | 0.74  | 0.477 |
| DFS   | -1.125 | -4.926,2.675  | -0.94 | 0.416 |
| OS    | -0.612 | -3.706,2.481  | -0.44 | 0.671 |

carcinoma and laryngeal carcinoma, which had little effect on OS in other solid tumors. We also found a small increase in DFS of colorectal carcinoma in high SHP2 expression compared with low one, rather than other solid tumors. It was more puzzling that the role of SHP2 in the prognosis of hepatocellular carcinoma. High SHP2 expression was associated with shorter DFS while early clinical stage in hepatocellular carcinoma. These further suggested that SHP2 might have more complex mechanisms in hepatocarcinogenesis and play complicated effects

on hepatocellular carcinoma. These results showed the dual role for SHP2 in tumorigenesis, which might be due to the different mechanism of SHP2 expression.

The activation of SHP2 by mutated EGFR is crucial for EGFR mutation driven lung adenocarcinoma [25]. That SHP2 and PDGFRα interacting with Dyn2 makes a valuable contribution in the growth and invasion of glioblastoma [26]. In breast cancer, SHP2 participates in tumor initiating cell maintenance and tumor growth by activating stemness-associated transcription factors and MAPK [27]. In prostate cancer, SHP2 promotes metastasis by enhancing epithelial mesenchymal transition [28]. SHP2 accelerates the growth and metastasis of HCC by coordinating the activation of Ras/Raf/Erk pathway and PI3-K/Akt/mTOR cascade [10]. Above these mechanisms explain why SHP2 is a poor prognostic marker of tumors. On the contrary, hepatocyte-specific SHP2 knock-out leads to the development of HCC in mice by activating Stat3, suggesting that SHP2 can inhibit tumor growth [29]. SHP2 inhibited CRC cell proliferation via STAT3 dephosphorylation [30]. In the process of liver tumorigenesis, SHP2 may act as a tumor promoter in vitro, but as a tumor suppressor in vivo [18], which may be why SHP2 play complicated effects on hepatocellular carcinoma. Perhaps because of these mechanisms, SHP2 shows a tumor suppressor gene. In addition, SHP2 is very important to maintain the immunosuppressive microenvironment by promoting the activation of M2 macrophages and inhibiting the activation of T cells [31]. Some SHP2 inhibitors, including allosteric inhibitors and enzyme inhibitors, have entered clinical trials, and some small molecular compounds have also displayed the latent capacity to restrain SHP2 [32]. Hence, this meta-analysis is of great value for guiding new targets of tumor treatment.

Egger's test did not show significant difference in clinical stage, DFS, and OS from the studies. That meant the non-existent of publication bias in clinical stage, DFS and OS, and these results were reliable.

However, there were some potential limitations in this study. First, there existed considerable heterogeneity in this study, which might be due to differences in cancer types, cell scoring strategies, research era and treatment strategies, etc. These limited us to obtain more comprehensive results. Second, since the prognosis of SHP2 seemed to vary greatly depending on the location of the tumor, the overall analysis of all types of cancers might depend highly on the relative proportions of each type of cancer. Be careful when interpreting the result. Third, although most of the data in the study were directly obtained, some studies only provided survival curves, leading to the deviation between the estimated and the actual statistical data. Detailed steps have been taken to minimize deviations. At last, the population included in the study was mainly from east Asia, not a good representation of the worldwide population.

In conclusion, this meta-analysis suggested the prognostic value of SHP2 might not equivalent in different tumors. Thus, the previous view that SHP2 invariably induced the proliferation of tumor was oversimplified. The difference of prognostic effect of SHP2 might be related to the different biological characteristics and diverse regulatory mechanism of specific tumor types. Go a step further to understand the effect of SHP2 in different human tumors will help to develop more accurate and effective immunotherapies.

## Supporting information

**S1 Checklist. PRISMA 2009 checklist.**
(DOC)

**S1 Table. Results of quality assessment using the Newcastle-Ottawa scale for included studies.**
(DOC)

## Author Contributions

**Conceptualization:** Jiupeng Zhou, Hui Guo, Quanli Dou.

**Data curation:** Jiupeng Zhou, Hui Guo, Quanli Dou.

**Formal analysis:** Yongfeng Zhang, Heng Liu.

**Funding acquisition:** Jiupeng Zhou.

**Investigation:** Quanli Dou.

**Methodology:** Yongfeng Zhang, Heng Liu.

**Resources:** Yongfeng Zhang.

**Software:** Heng Liu.

**Writing – original draft:** Jiupeng Zhou.

**Writing – review & editing:** Jiupeng Zhou, Hui Guo, Quanli Dou.

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
