## [Decision Letter · Decision Letter 0]

30 Dec 2021

PONE-D-21-25830

Prognostic significance of SHP2 (PTPN11) expression in solid tumors: A meta-analysis

PLOS ONE

Dear Dr. Zhou,

Thank you for submitting your manuscript to PLOS ONE. After careful consideration, we feel that it has merit but does not fully meet PLOS ONE’s publication criteria as it currently stands. Therefore, we invite you to submit a revised version of the manuscript that addresses the points raised during the review process.

Please address the following two comments from the reviewer before accepting for the publication.

1.       To check for grammar correction

2.       Author can focus on the explanation part for the data utilized for Subgroup Meta-analyses and publication bias.

We look forward to receiving your revised manuscript.

Kind regards,

Johnson Rajasingh, Ph.D, HCLD

Academic Editor

PLOS ONE

2. Please confirm that you have included all items recommended in the PRISMA checklist including the full electronic search strategy used to identify studies with all search terms and limits for at least one database.

Reviewers' comments:

Reviewer's Responses to Questions

**Comments to the Author**

1. Is the manuscript technically sound, and do the data support the conclusions?

Reviewer #1: Yes

2. Has the statistical analysis been performed appropriately and rigorously? 

Reviewer #1: Yes

3. Have the authors made all data underlying the findings in their manuscript fully available?

Reviewer #1: Yes

4. Is the manuscript presented in an intelligible fashion and written in standard English?

Reviewer #1: Yes

5. Review Comments to the Author

Reviewer #1: The article seems to have a sound methodology and the author explained well with the meta-analysis part to show the prognostic value of SHP2. I suggest the author with minor suggestion to be done below

1. To check for grammar correction

2. Author can focus on the explanation part for the data utilized for Subgroup Meta-analyses and publication bias.

6. PLOS authors have the option to publish the peer review history of their article (what does this mean?). If published, this will include your full peer review and any attached files.

Reviewer #1: **Yes: **Dr. Magesh R

---

## [Author Response · Author response to Decision Letter 0]

3 Jan 2022

1. To check for grammar correction 

Response: Grammar correction has been performed.

2. Author can focus on the explanation part for the data utilized for Subgroup Meta-analyses and publication bias.

Response: The explanation for the data has been supplemented.

These studies provided almost all the available evidences worldwide on the role of SHP2 in the prognosis of solid tumors. A advantage of bringing together worldwide evidences on the association between SHP2 and the prognosis of solid tumors was that there were large numbers of cases to assess reliably whether the association varies by tumour subtype. We found that high SHP2 expression was associated with shorter OS in two subtypes of solid tumors of pancreatic carcinoma and laryngeal carcinoma, which had little effect on OS in other solid tumors. We also found a small increase in DFS of colorectal carcinoma in high SHP2 expression compared with low one, rather than other solid tumors. It was more puzzling that the role of SHP2 in the prognosis of hepatocellular carcinoma. High SHP2 expression was associated with shorter DFS while early clinical stage in hepatocellular carcinoma. These further suggested that SHP2 might have more complex mechanisms in hepatocarcinogenesis and play complicated effects on hepatocellular carcinoma.

Egger’s test did not show significant difference in clinical stage, DFS, and OS from the studies. That meant the non-existent of publication bias in clinical stage, DFS and OS, and these results were reliable.

---

## [Editor Report · Decision Letter 1]

10 Jan 2022

Prognostic significance of SHP2 (PTPN11) expression in solid tumors: A meta-analysis

PONE-D-21-25830R1

Dear Dr. Zhou,

We’re pleased to inform you that your manuscript has been judged scientifically suitable for publication and will be formally accepted for publication once it meets all outstanding technical requirements.

Kind regards,

Johnson Rajasingh, Ph.D, HCLD

Academic Editor

PLOS ONE
---

## [Editor Report · Acceptance letter]

12 Jan 2022

PONE-D-21-25830R1 

Prognostic significance of SHP2 (PTPN11) expression in solid tumors: A meta-analysis 

Dear Dr. Zhou:

I'm pleased to inform you that your manuscript has been deemed suitable for publication in PLOS ONE. Congratulations! Your manuscript is now with our production department. 

Kind regards, 

on behalf of

Dr. Johnson Rajasingh 

Academic Editor

PLOS ONE